# Acoustic levitation of pollen and visualisation of hygroscopic behaviour

Sophie A. Mills[1,2], Adam Milsom[1], Christian Pfrang[1,2,3], A. Rob MacKenzie[1,2], Francis D. Pope[1,2]

[1]School of Geography, Earth and Environmental Sciences, University of Birmingham, Birmingham, B15 2TT, UK
[2]Birmingham Institute of Forest Research, University of Birmingham, Birmingham, B15 2TT, UK
[3]Department of Meteorology, University of Reading, Whiteknights, Earley Gate, Reading, RG6 6BB, UK

*Correspondence to*: Francis D. Pope (f.pope@bham.ac.uk)

**Abstract.** Pollen are hygroscopic and so have the potential to act as cloud condensation nuclei (CCN) in the atmosphere. This could have yet uncertain implications for cloud processes and climate, as well as plant biodiversity and human health. Previous
studies have investigated the hygroscopic swelling of pollen, linked to CCN activity by the κ-Köhler theory, using methods that follow observed mass increase by electrodynamic balance (EDB) or vapour sorption analyser. This study uses an acoustic levitator to levitate pollen grains in the true aerosol phase and uses a macroscope to image the pollen to investigate hygroscopic behaviour when relative humidity (RH) is changed. Two pollen species were studied in this work: *Lilium orientalis* (oriental lily) and *Populus deltoides* (eastern cottonwood). Both species were successfully levitated, however, the smaller *Populus*
*deltoides* showed greater instability throughout experiments. The quality of images taken by the macroscope, and thus calculations of pollen area and aspect ratio, varied significantly and were sensitive to lighting conditions, as well as levitated pollen grain movement and orientation. Experiments with surface-fixed pollen grains were also conducted. They showed evidence that pollen hygroscopic swelling could be observed by the macroscope. The produced results were comparable with previously reported mass increase values. Although less accurate than methods that measure mass changes, the acoustic
levitator and macroscope setup offer an attractive alternative by virtue of being commercial-off-the-shelf, low-cost, and versatile. A key advantage of this method is that it is possible to visually observe particle shape dynamics under varying environmental conditions.

## 1 Introduction

Primary biological aerosol particles (PBAPs, or bioaerosols) are airborne particles derived from biological organisms (Després
et al., 2012) and come from a huge variety of sources including plants, animals and microorganisms. They comprise a significant proportion of total atmospheric aerosol mass, calculated to be around 20-30% in urban/rural regions (Matthias-Maser and Jaenicke, 1995; Matthias-Maser et al., 2000) and 10% and 20% in marine and remote continental regions, respectively, (Gruber et al., 1998) for PBAPs greater than 0.2 μm in diameter. As with other aerosol particles, PBAPs are important in the Earth system and while in the atmosphere they undergo transportation, chemical and physical transformations,
and cloud processing (Frohlich-Nowoisky et al., 2016; Després et al., 2012). PBAPs have been found to constitute up to 25%

of insoluble aerosol particle number concentrations with diameters greater than 0.2 μm in cloud water (Matthias-Maser et al., 2000) and between 9-50% in rainwater (Diehl et al., 2001), so their potential to influence cloud processes and affect precipitation cannot be neglected (Andreae & Rosenfield, 2008).

Meanwhile, PBAPs play an important biological role, with pollen being responsible for the spread of plant species and colonisation of new ecosystems. Furthermore, many have direct relevance to human health. Pollen are responsible for allergic reactions such as hay fever and asthma. The size and aerodynamic ability of these particles affect their average travel distance and dynamics, so understanding of the physical processes affecting these particles is essential for accurate modelling, spatial concentration and personal exposure estimations.

Pollen particles are among the largest in physical size of PBAPs, generally measuring between 10 and 100 μm in diameter (see, e.g., Després et al., 2012; Mills et al., 2023), yet vary considerably in size and shape between and even within species. While they are not as numerous in the atmosphere as other PBAPs such as bacteria and fungal spores, pollen particles contribute significantly to the global emissions mass, estimated to be around 47-84 Tg yr$^{-1}$ (Després et al., 2012). Most airborne pollen comes from anemophilous trees, shrubs and grasses, whose pollen grains have evolved to be dispersed by wind and are therefore smaller in size, typically between 20-45 μm (Steiner et al., 2015). Such pollen species have characteristic seasons of pollen release so airborne pollen concentrations fluctuate throughout the year, and across the globe depending on local vegetation species.

Airborne pollen concentrations are measured to be anywhere between 10-1000 grains m$^{-3}$ (Després et al., 2012), but can reach $10^4$-$10^6$ grains m$^{-3}$, particularly in the Northern hemisphere spring months of April and May when many deciduous trees peak in pollen emissions (Steiner et al., 2015; Gregory, 1978). Pollen grains have been shown to reach altitudes of 3 km and above in the atmosphere (Diehl et al., 2001) and travel horizontal distances of up to ~$10^3$ km (Sofiev et al., 2006; Hjelmroos, 1992; Sack, 1949). Individual small pollen grains can reside in the atmosphere for a few (generally 2-4) days (Sofiev et al., 2006), though pollen seasons can persist with high average concentrations (~$10^4$ grains m$^{-3}$) over many months, fluctuating with locality, season, time of day and meteorology (Gregory, 1928).

When present in the atmosphere, pollen grains have the potential to act as cloud condensation nuclei (CCN) - atmospheric particles which provide a surface for water vapour to condense on and cloud droplets to grow. A prerequisite for active CCN particles is that their surface is wettable and they are hygroscopic, and pollen grains have been demonstrated to adsorb moisture from the air and swell under high humidity, displaying moderate hygroscopicity (Pope, 2010; Griffiths et al., 2012; Chen et al., 2019). This has been attributed to the numerous hydrophilic hydroxyl (OH) groups of organic compounds in pollen (Tang et al., 2019). Note that when we refer to hygroscopicity of pollen it should more correctly be 'apparent hygroscopicity', since pollen grains are a mixture of soluble and insoluble components.

Though relatively small in number concentration (i.e. generally 10 m$^3$ compared to the $10^3$-$10^4$ m$^3$ number concentrations of fungal spores, bacteria and viral particles – see, e.g., Després et al.), the large size of pollen grains mean that they can act as coalescence embryos, also known as giant cloud condensation nuclei (GCCN). These GCCNs have been shown within atmospheric models to have a disproportionate contribution to the development of precipitation within clouds (Johnson, 1982;

Cotton and Yueter, 2009). They can nucleate cloud droplets at lower supersaturations than other smaller aerosols and facilitate precipitation more rapidly (Després et al., 2012; Pope, 2010; Posselt & Lohmann, 2008), and particularly affect precipitation in clouds of high droplet concentration, i.e. polluted as opposed to pristine clouds, even at concentrations as low as $10^{-3}$ cm$^{-3}$ (Möhler et al., 2007). Additionally, pollen grains and their extracts have shown potential to act as ice nuclei (IN) and induce heterogenous ice formation in the atmosphere (Diehl et al., 2001 & 2002; Dreischmeier et al., 2017; Pummer et al., 2012; Tong

et al., 2015). While there have been discussions in the literature postulating the significance pollen may have for atmospheric cloud processes and climate, there is need for further experiments to measure and better understand the hygroscopic behaviour of pollen and other biological particles (Möhler et al., 2007; Després et al., 2012; Sun & Ariya, 2006). Pollen hygroscopicity may also have important implications for airborne transmission to new areas, and thus the colonisation and biodiversity of various plant species. Additionally, it is important to consider for human health, particularly since the release of respirable-

sized subpollen particles from pollen grains can be modulated by humidity conditions (Matthews et al., 2023) and stimulated by events such as thunderstorms (Hughes et al., 2020; Cecchi et al., 2021).

Previous studies have attempted to quantify the hygroscopic water uptake of a variety of different types of pollen under varying RH (relative humidity) conditions by measuring the corresponding mass change. Table 1 presents a summary of the relevant findings of each study, including the pollen species studied, the method employed, the RH interval and corresponding mass

increase. Some of the studies also report corresponding κ values, derived from the κ-Köhler theory (Petters and Kreidenweis, 2007) which links the hygroscopic size change of a particle with CCN activity, based on Köhler theory (Köhler, 1936). This κ value is determined from fitted hygroscopic growth factor data and quantifies hygroscopic growth, and thus CCN potential (Petters and Kreidenweis, 2007).

A particle is non-hygroscopic at κ = 0 and increases in hygroscopicity and CCN potential with increasing κ value. This κ value

is generally found in the range 0.1-0.9 for atmospheric particles, with an upper limit of approximately 1.4 for sodium chloride (sea salt). However, the range for 'slightly to very' hygroscopic organic species is typically found towards the lower end of the scale between 0.01 and 0.5 (Petters and Kreidenweis, 2007). Previous studies have calculated κ values of various pollen species under subsaturated RH to be between 0.05-0.1 which, while suggesting moderate-low hygroscopicity, is not negligible since there is significant CCN potential at critical supersaturations considering their large size (Pope, 2010; Tang et al.,

2019).Pollen grains are complex biological structures with varying sizes, shapes and surface features across species. Previous methods have focused on the measurement of mass increase with hydration and have not considered the simultaneous changes that may occur in volume and density. For example, Božič and Šiber (2022) explore the mechanics of pollen grain unfolding, swelling and bursting under humidity changes, which is largely dictated by the soft apertures, or pores, found on the exine (outer surface). The number and size of pores can vary across different pollen taxa but the mechanics involving them are not

well understood. Alongside modelling studies such as Božič and Šiber (2022), empirical studies which shed light on the visual changes that occur on the pollen grain surface would bring value. With these considered, we may better understand the complexities of pollen hygroscopic behaviour and how it should be modelled in the atmosphere.

This study presents a novel approach, using an acoustic levitator to suspend pollen and a macroscope camera to capture images of the grains as they undergo changes in RH. As opposed to following the hygroscopic swelling of the pollen grains by the change in mass, this approach investigates the feasibility of following the hygroscopic behaviour visually by the change in pollen size.

**Table 1: Previous studies investigating the hygroscopic behaviour of pollen, including the pollen species used for each study, the apparatus used for each method, the RH interval studied, the corresponding mass increase percentage observed (the average or range across all pollen species for each study is generally cited), and the corresponding κ value from the κ-Köhler theory where it has been calculated.**

| Reference | Pollen species | Method | RH interval | % mass increase | κ value |
|---|---|---|---|---|---|
| **Diehl et al. 2001** | *Betula alba*, *Alnus incana*, *Populus nigra*, *Quercus rubra*, *Pinus sylvestris*, *Picea alba*, *Agrostis alba*, *Dactylis glomerata*, *Poa pratensis*, *Secale cereale* | Analytical balance | Dry-73%<br><br><br><br>Dry-95% | 3-16%<br><br><br><br>~100-300% | - |
| **Pope 2010** | *Narcissus spp.*, *Betula occidentalis*, *Salix caprea*, *Juglans nigra* | Electrodynamic balance | 0-75% | 16% | 0.05-0.10 |
| **Griffiths et al. 2012** | *Secale cerale*, *Kochia scoparia*, *Artemisia tridentata*, *Iva xantifolia* | Electrodynamic balance | 0-75% | 18% | 0.05-0.22 |
| **Bunderson & Levetin 2015** | *Juniperus ashei*, *J. monosperma*, *J. pinchotii* | Analytical balance | 20-97% | 21-25% | - |
| **Tang et al. 2019** | *Carya illinoinensis* (pecan) | Vapour sorption analyser | 0-30%;<br>0-60%;<br>0-90%<br>0-95% | 2.3%<br>6.4%<br>30.3%<br>~72% | 0.034-0.067 |
|  | *Populus tremuloides, Populus deltoides, ragweed, corn, paper mulberry* |  | 0-90% | 29-48% |  |

| Chen et al. 2019 | *Pinus massoniana, Pinus tabuliformis, Pinus armandii, Pinus taiwanesis, Pinus bungeana, Typha angustifolia, Pyrus sp., Amygdalus persica, Malus pumila, Prunus salicina, Brassica campestris* | Vapour sorption analyser | 0-90% | 33-43% | 0.036-0.048 |

## 2 Methods

### 2.1 Pollen samples

Two different types of pollen were used in this study: *Lilium orientalis* (oriental lily) and *Populus deltoides* (eastern cottonwood). The *Lilium orientalis* pollen was collected from fresh lily flowers bought from a UK Sainsbury's supermarket as the anthers ripened. These flowers' pollen is not anemophilous, it is usually insect-pollinated. However, the large grain size made it a good candidate for initial experiments with the acoustic levitator, since larger particles are easier to levitate. *Lilium* pollen grains are generally large and have an ellipsoidal shape with the polar axis generally ranging between 70-120 μm while the equatorial axis is often around 30-70 μm, i.e. with an approximate polar-equatorial ratio of 2:1 (Du et al., 2014).

The *Populus deltoides* pollen was commercially sourced from Merck (Sigma-Aldrich) and is smaller than the *Lilium orientalis* pollen, with optical diameter generally between 23-34 μm as quoted by Rajora (1989). It was used in this study to represent common anemophilous tree pollen species, such as birch and oak, which are generally between 10-40 μm in size and generally more spheroidal in shape (polar-equatorial ratio between 0.88-1.14, as defined by Punt et al., 2006). *Populus deltoides* pollen has a somewhat triangular polar-view outline with usually three apertures like other common tree pollen species including birch and oak.

Pollen grains from the samples used in this study (see Fig. S1 in the SI for microscope images) were measured using a Nikon SPZ1000 stereo microscope equipped with a Nikon DS-Fi1 camera and controlled by NIS-Elements acquisition software at the Birmingham Advanced Light Microscopy (BALM) facility at the University of Birmingham. The *Lilium orientalis* pollen from fresh flowers was measured to have polar and equatorial diameters ranging between 92-122 μm and 43-67 μm and means of 108.6 μm and 50.9 μm, respectively. The *Populus deltoides* pollen used for this study was measured to have diameters across multiple axes ranging between 18 and 32 μm, with a mean of 24.8 μm. There is however considerable variation in size and shape across even pollen grains of the same taxa since these are gametes produced from living organisms with complex biochemistry and physiology.

## 2.2 Acoustic levitation chamber apparatus

The pollen grains were levitated using a modified commercial acoustic levitator (tec5, Oberursel, Germany) in which particles are suspended by the force of a standing acoustic wave between a transducer and opposing reflector. The application of acoustic levitation has progressed significantly in recent years. For example, acoustic levitation has been coupled with techniques such as Raman spectroscopy (Milsom et al., 2021; Pfrang et al., 2017) and X-ray scattering experiments (Milsom et al., 2021; Milsom et al., 2022; Pfrang et al., 2017; Seddon et al., 2016). Acoustic levitation presents opportunities to study substances in

contactless and container-less conditions, i.e. in the true aerosol phase. It has become an opportune, commercially available and low-cost alternative, as opposed to other methods, such as the electrodynamic balance (EDB), and demonstrates huge versatility in being able to levitate a wide variety of objects (Marzo et al., 2017; Andrade et al., 2018).

The setup for this study was based on that previously described by Milsom et al. (2021) and is illustrated in Fig. 1. The levitator operated at a fixed frequency (100 kHz) with variable power (0.65-5.00 W) and was initially deployed for studies of a wide

range of self-assembled materials (e.g. Seddon et al., 2016; Pfrang et al., 2017). The concave reflector, parallel above the transducer, was fitted with a micrometre screw to facilitate adjustment relative to the transducer. The reflector-transducer distance was adjusted accordingly to levitate particles, but generally kept in the range of 20-30 mm, and was kept constant for the duration of each experiment. A custom-built 3D printed chamber enclosed the levitator and was sealed from ambient air outside except for access ports for the RH sensor and tubing to deliver humidified air. The system for monitoring and

controlling RH in real-time was custom-made, using a Raspberry Pi connected to an air pump and an Arduino connected to a DHT22 RH sensor (2–5% RH accuracy between 0-100% RH). 'Dry' (lab ambient) air was pumped through tubing which split into two flows: one was bubbled through water to supply humidified air while the other dry air flow was passed through a manual flow meter to control the dry relative to wet flow. The two flows were re-joined and delivered into the chamber, where the RH sensor was also positioned, and the flow was adjusted accordingly to cover the experiment humidity range. A

macroscope with a digital camera attached (Leica MC190 HD) was positioned in front of one of the levitation chamber's transparent microscope slide windows to take snapshots throughout the experiments. A photograph of the acoustic levitator as used in this study can be found in Fig. S1 in the Supporting Information.

**Figure 1: Diagram illustrating setup of acoustic levitation chamber, humidification system and macroscope for experiments**
**investigating the hygroscopic behaviour of pollen.**

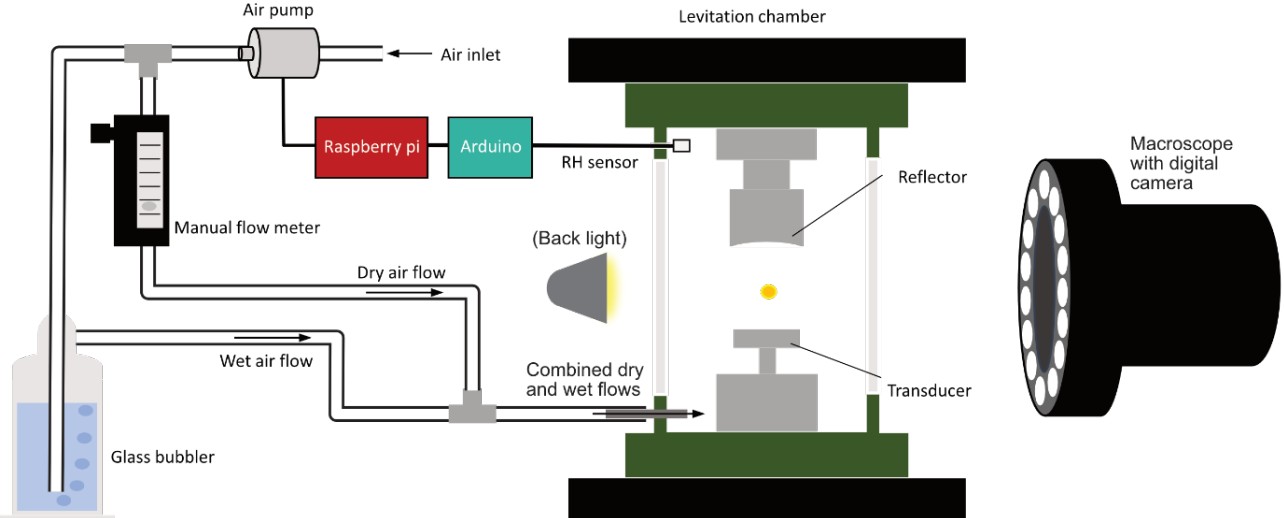

## 2.3 Hygroscopicity experiments

For instances where single pollen grains were levitated, these grains were first isolated on a slide under a compound microscope before being introduced to the levitation chamber via pin-tip. This was to make sure that, where possible, single grains were observed. Once levitated, the chamber windows were sealed using tape, lighting applied to illuminate the pollen grain, and the focus of the macroscope adjusted onto the pollen grain. Lighting was directed either from the front, by the lights encircling the macroscope camera lens, or from the back, by a Cole-Parmer Fiber Optic Illuminator (Model 41720), though back-lighting was reverted to as default for reasons discussed in section 3. Humidity was increased by increments of generally 5 or 10% RH from ambient room RH up to the maximum RH that could be reached and at each increment 5 snapshots, or in some cases more, were captured. Once maximum RH was reached, the process was repeated for decreasing RH increments back to ambient RH.

Surface-fixed experiments were conducted using a similar RH stepwise procedure, except that pollen grains were placed on a microscope cover slip which was attached to the inside of the levitation chamber window facing the macroscope. These experiments did not investigate the true aerosol phase of airborne pollen grains but ensured the pollen grains were fixed and that the orientation was constant throughout the experiment.

## 2.4 Image and data processing

The images captured by the macroscope camera during each experiment were taken and processed by a bespoke program written using Python 3.9 for the specific purposes of this experiment. This Python program applies a binary threshold, performs Canny edge detection and identifies object contours (i.e. the pollen grain outlines) within the selected image range (see Fig. S2 in the SI for a diagram of the full methodology of this process). An area measurement is recorded for the largest contour found (also checking visually that this is the pollen grain) and an ellipse is fitted to the same contour from which 'polar' and

'equatorial' diameters are extracted in pixel units. Due to the image contrast setting being adjusted as each grain was levitated, the threshold value in the program had to be changed accordingly, therefore, it is consistent only within each experiment run. The raw measurements calculated by the image processing program were processed and visualised using Python 3.9.7 in Jupyter Notebook (Anaconda Distribution) with package versions Numpy 1.24.1, Pandas 1.5.3, Matplotlib 3.6.3 and Seaborn 0.12.2. Pixel area measurements have been converted into average area increase ratios (averaged over 5 repeat snapshot images) relative to the initial size measured at ambient RH (for each experiment), defined in Eq. (1):

$$Area\ ratio\ = \frac{area\ within\ contour\ at\ given\ RH\ (pixels)}{area\ at\ initial\ RH\ (pixels)} \tag{1}$$

The ellipse-fitted diameter measurements do not necessarily correspond with 'polar' and 'equatorial' axes defined by general pollen terminology, as this cannot be determined with certainty from the images. However, we used the ellipse diameter measurements to calculate an average aspect ratio to evaluate change in shape. The aspect ratio was defined by the following Eq. (2):

$$Aspect\ ratio\ = \frac{smaller\ \text{(equatorial)}\ diameter}{larger\ \text{(polar)}\ diameter} \tag{2}$$

This resulted in values between 0 and 1, with higher values implying more circularity (and thus likely higher sphericity of the whole pollen grain as well). The standard deviation was calculated across the 5 repeats for each averaged value of area and aspect ratios and is presented as error bars in the figures in Results and Discussion sections 3.2 and 3.3.

To facilitate meaningful comparison between experiment results, we fitted a simple least squares linear regression to each pollen grain experiment using the Scikit-learn (version 1.2.1) LinearRegression model. These models fitted area and aspect ratios against RH for the direction of increasing RH only in each case. The linear model is not a good model to describe the behaviour of hygroscopic size change on an incremental level. It was the most parsimonious model to derive the size and shape changes between a consistent RH range while taking all data points fairly into account (i.e. not just taking the start and finish values which could be anomalous). From the fitted models for each instance, we predicted the area and aspect ratio changes between 70 and 95% RH. (These values were chosen as they generally lay within the experimentally measured RH ranges for all instances.) The results were summarised as a box and whisker plot for each pollen type and experiment setup (i.e. surface-fixed and levitated). A paired, two-sided t-test (using SciPy 1.10.0) was also performed for each experiment group between data points estimated by the linear model at 70 and 95% RH, respectively, to test if the observed difference was significant.

## 3 Results and Discussion

### 3.1 Acoustic levitation and visual imaging of pollen grains

Both *Lilium orientalis* and *Populus deltoides* pollen were successfully levitated in the acoustic levitator, as clusters of multiple pollen grains and single grains, though this was achieved with varying ease and stability. Adjustment of the transducer frequency and transducer-reflector distance were necessary at times when the pollen grains were unstable to the point that snapshot quality was compromised by the erratic motion of the levitated particle.

The stability of an acoustically levitated particle depends on its position relative to nodes created by the pattern of standing waves between transducer and reflector. The acoustic waves produce an upward pressure that counteracts the force of gravity, while levitated objects are most stable when sat at node points where the wave movement and pressure is at a local minimum. The node positions depend on both the frequency of the sound waves being produced and the transducer-reflector distance. The pollen grains may not always manage to settle at a node point and therefore can be prone to instability – moving erratically, spinning, and sometimes dropping out of the trap altogether.

This instability occurred more frequently for the smaller *Populus deltoides* pollen than *Lilium orientalis* pollen, and for single grains as opposed to clusters of multiple grains. These particles, being lighter, were likely more sensitive to fluctuations in their position relative to the acoustic field delicately balancing them, and therefore were more easily thrown out of balance if not securely settled at a node point. Asymmetrical or irregular clusters of pollen grains such as those shown in Fig. 2 were also less stable, likely due to the juxtaposition of forces exerted by different areas of the acoustic field across the irregular structure, and this structure was frequently re-organised. Thus, for the following hygroscopicity experiments we always attempted to isolate and levitate single grains to remove the variability of cluster structure.

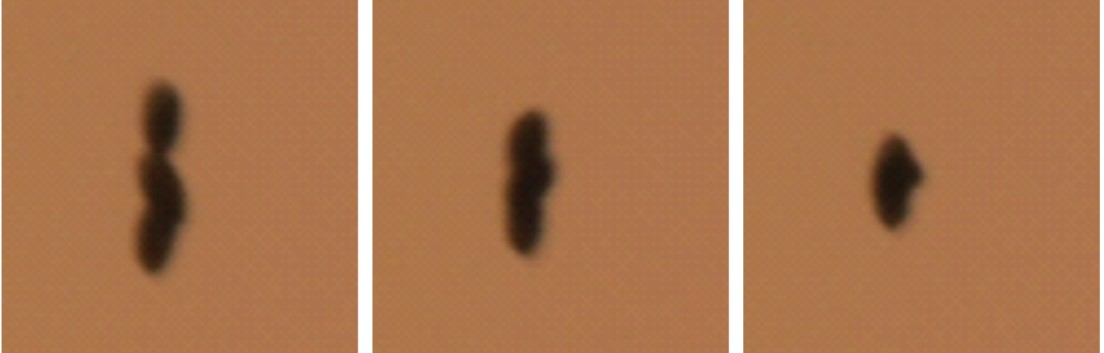

**Figure 2: Example of multiple *Lilium orientalis* pollen grains being levitated at once but experiencing instability and fluctuation in structure and orientation. These photos are a representative few of many similar snapshots that were captured in the space of a few minutes.**

Table 2 shows representative snapshots taken for each pollen type and experiment setup to show general image quality and the variation observed among them. Further images showing examples of the pollen grains across the relative humidity range, as well as the program-fitted contour and ellipse used for measurements, can be found in Figures S4 and S5 in the SI. All experiment images used in this study, as well as relevant code and datasets, can be found in the data repository in the Data Availability section.

The macroscope camera contrast was adjusted for each experiment to get the best image with most crisp and easily definable edges; however, this was achieved to varying success for each case. From initial experiments, back-lighting was chosen as a default to proceed with measurements as it resulted in a generally crisper and more easily definable silhouette. Importantly, it also produced a cleaner background devoid of other features that would interfere with the Canny edge detection and contour identification. (Front-lighting resulted in a less-pristine background due to other features picked up from the glass chamber window.)

In the following results, we include for comparison an example where two pollen grains were observed under front-lit conditions. After post-experiment analysis, it could be argued that they achieved more consistent results than the equivalent back-lit experiment. The decision to proceed with back-lighting was made based on observations at the time for the reasons above, in particular with the automated method to distinguish pollen grains from the image background in mind. However, the method could be refined to overcome interference of background features and it may be that front-lit experiments are worth testing again in the future, as this may produce more consistent images that vary less with light brightness and contrast settings.

**Table 2. Representative snapshots taken for each pollen type and experiment setup showing the image quality achieved as well as the variation across different instances. Further snapshots showing example pollen grains across the relative humidity range can be found in Figures S4 and S5 in the SI.**

| Pollen type | Levitated or surface-fixed | Back or front-lit | Example pictures |
|---|---|---|---|
| *Lilium orientalis* | Surface-fixed | Back-lit |  |
| *Lilium orientalis* | Surface-fixed | Front-lit |  |

| | | | |
|---|---|---|---|
| *Lilium orientalis* | Levitated | Back-lit |  |
| *Populus deltoides* | Surface-fixed | Back-lit |  |
| *Populus deltoides* | Levitated | Back-lit |  |

## 3.2 Surface-fixed pollen hygroscopicity experiments

### *Lilium orientalis*

The results for the *Lilium orientalis* pollen fixed to the levitation chamber window and under back-lit conditions are shown in Fig. 3. One pollen grain, the blue line with circle markers in Fig. 3 (further images presented in Fig. S4 in the SI), increased in silhouette area to just over 20% between 70-96% RH. Additionally, both directions of increasing and decreasing RH showed a similar trend without hysteresis. Another grain observed under the same conditions (orange with upright triangle in Fig. 3) showed some evidence of increasing by around 2% but ultimately produced an inconclusive trend for increasing RH. Meanwhile the same grain showed a little decrease in area with decreasing RH but generally only by a few percent. The two pollen grains were different in shape, with different starting aspect ratios of 0.79 and 0.47 - i.e. one more circular and the other more ellipsoidal - for the first and second grain, respectively. The aspect ratio for the first decreased with increasing RH to 0.76 and continued to decrease further down to 0.67 with decreasing RH which seems contradicting. Meanwhile the second grain's aspect ratio increased up to 0.54, then decreased down to 0.45 with increasing and decreasing RH correspondingly.

**Figure 3: Visualised results for the surface-fixed Lilium orientalis pollen grains undergoing incremental RH changes with error bars. A & B: Area ratio against increasing and decreasing RH respectively. C & D: Aspect ratio against increasing and decreasing RH, respectively. The colours correspond with the respective pollen grain images shown beside them and lines between the discrete measurements have been included for visual aid.**

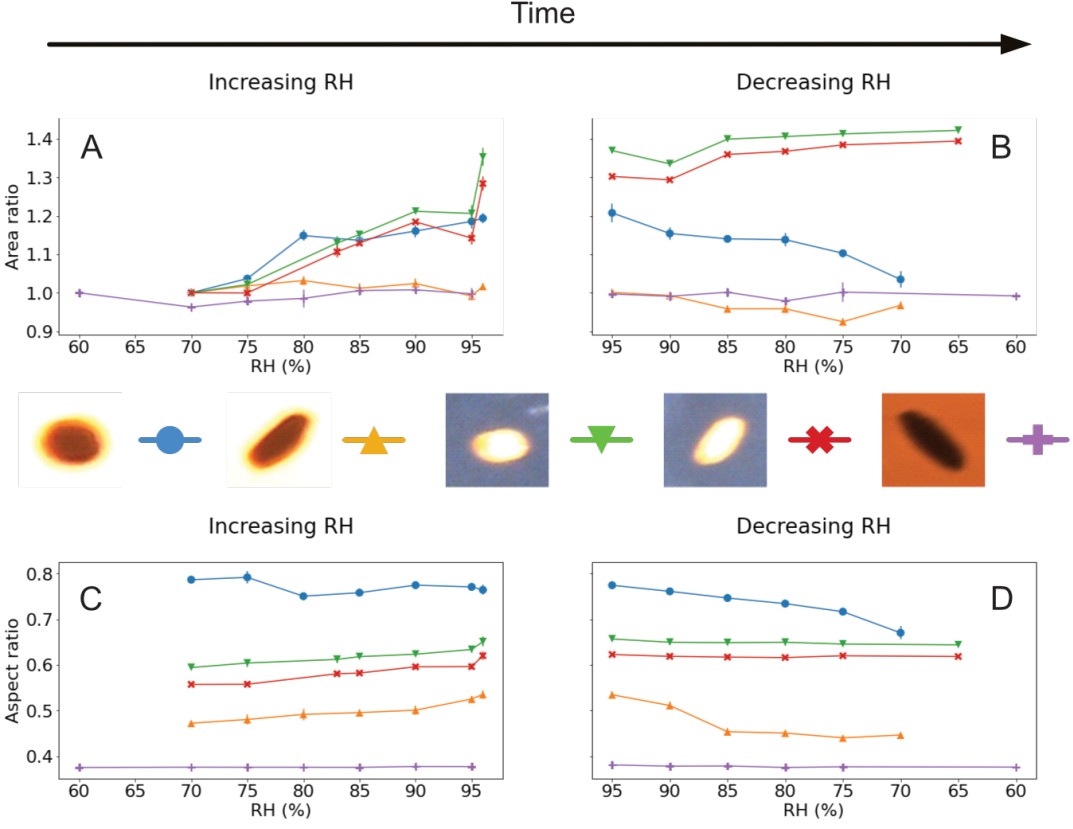

275

The same pollen grains were observed under front-lighting, giving a considerably different picture as can be seen in Table 2. Here, on the contrary, the two grains behaved very similarly (green with upside-down triangles and red crossed in Fig. 3). The first grain appeared more elongated and similar the second grain, and each had an initial aspect ratio of 0.59 and 0.56, respectively. Both aspect ratios increased with increasing RH (to 0.65 and 0.62, respectively) but showed no significant change for decreasing RH. Meanwhile, both front-lit grains increased in area with increasing RH by 35 and 28%. They increased even further with decreasing RH, up to 42 and 39%, before starting to decrease a little by the end of the experiment.

A different grain tested once again in a back-lit experiment (purple plus signs in Fig. 3) showed no increase or significant decrease in area with change in RH, too. This grain started with a aspect ratio of 0.38 and this too did not fluctuate significantly throughout the experiment.

While these results show some evidence of the hygroscopic size increase we had expected with increasing RH, they also suggest apparent inaccuracy and lack of consistency which must be considered. This variability may be due to limitations of the method in terms of imaging capabilities. Meanwhile there is possibly more convincing evidence of increased circularity –

and, by assumption, increased sphericity - of grains as they undergo hygroscopic morphology changes under increased RH. In particular, this trend appears more evident for grains that are initially less circular.

Importantly, these examples show the sensitivity of the experiment to lighting conditions, including contrast settings, of the image taken. The lighting conditions influence the visible pollen grain outline and, consequently, the measurements calculated by the image processing program from these snapshots. It may not be possible to have a one-setting-fits-all for image contrast and it is in turn difficult to relate the observed outline to an absolute value for the pollen grain size. However, it is imperative that the lighting conditions and contrast settings are kept constant throughout each experiment run, as was done for all experiments reported here. This ensures that the change in size and shape of a particular grain silhouette within an experiment run is comparable and meaningful.

*Populus deltoides*

**Figure 4: Visualised results for the surface-fixed Populus deltoides pollen grains undergoing incremental RH changes with error bars. A: Area ratio against increasing RH. B: Aspect ratio against increasing RH. The colours correspond with the respective pollen grain images shown beside them and lines between the discrete measurements have been included for visual aid. Decreasing RH was not measured in this instance.**

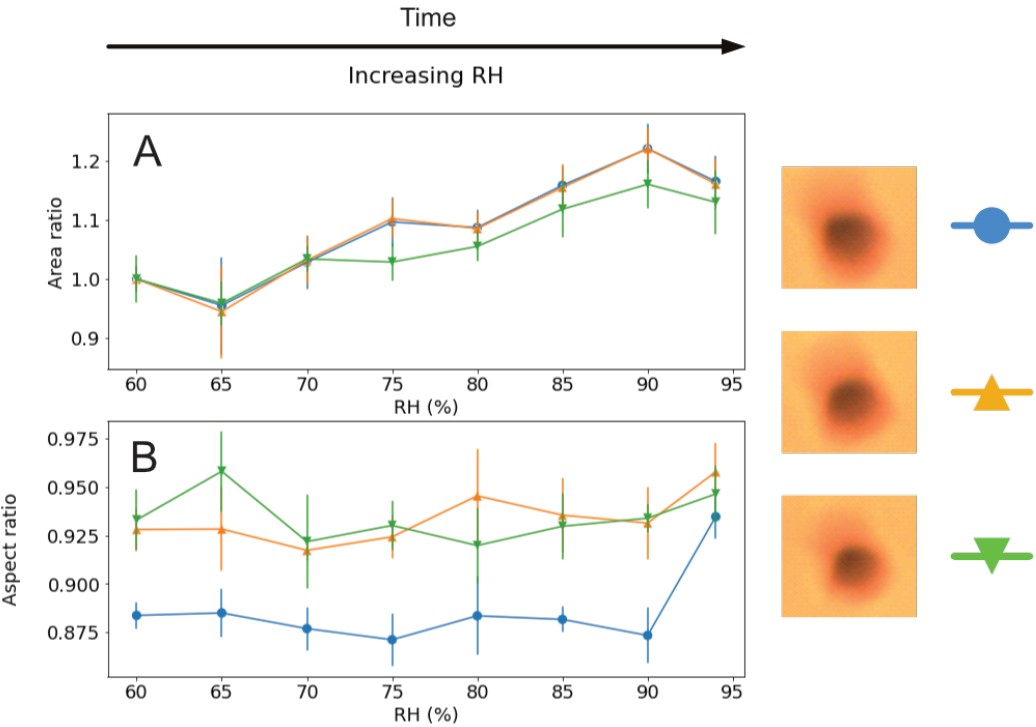

For the surface-fixed *Populus deltoides* pollen experiments (all back-lit), shown in Fig. 4, three grains were observed simultaneously under the same conditions and all showed similar results. That is, all three grains exhibited a steady increase in area by 13-22% between 60 and 94% RH. Decreasing RH was not recorded for this experiment. The aspect ratios, which were all initially near-circular (aspect ratios between 0.87-0.94), showed some signs of slight increase in circularity to a range of 0.93-0.95 (the grain with the lowest initial ratio increasing the most) yet the change may not be considered conclusive. These results show evidence of a hygroscopic size increase for *Populus deltoides* pollen with increasing RH. While there is some evidence of increased grain circularity, these grains are already near-spherical to start with and so a less significant ratio change is to be expected. Nevertheless, since these grains were all simultaneously under the same conditions it would be necessary to conduct more experiment instances before making conclusive assertions from these results.

### 3.3 Levitated hygroscopicity experiments

*Lilium orientalis*

**Figure 5: Visualised results for the levitated Lilium orientalis pollen grains undergoing incremental RH changes with error bars. A & B: Area ratio against increasing and decreasing RH, respectively. C & D: Aspect ratio against increasing and decreasing RH respectively. The colours correspond with the respective pollen grain images shown beside them and lines between the discrete measurements have been included for visual aid.**

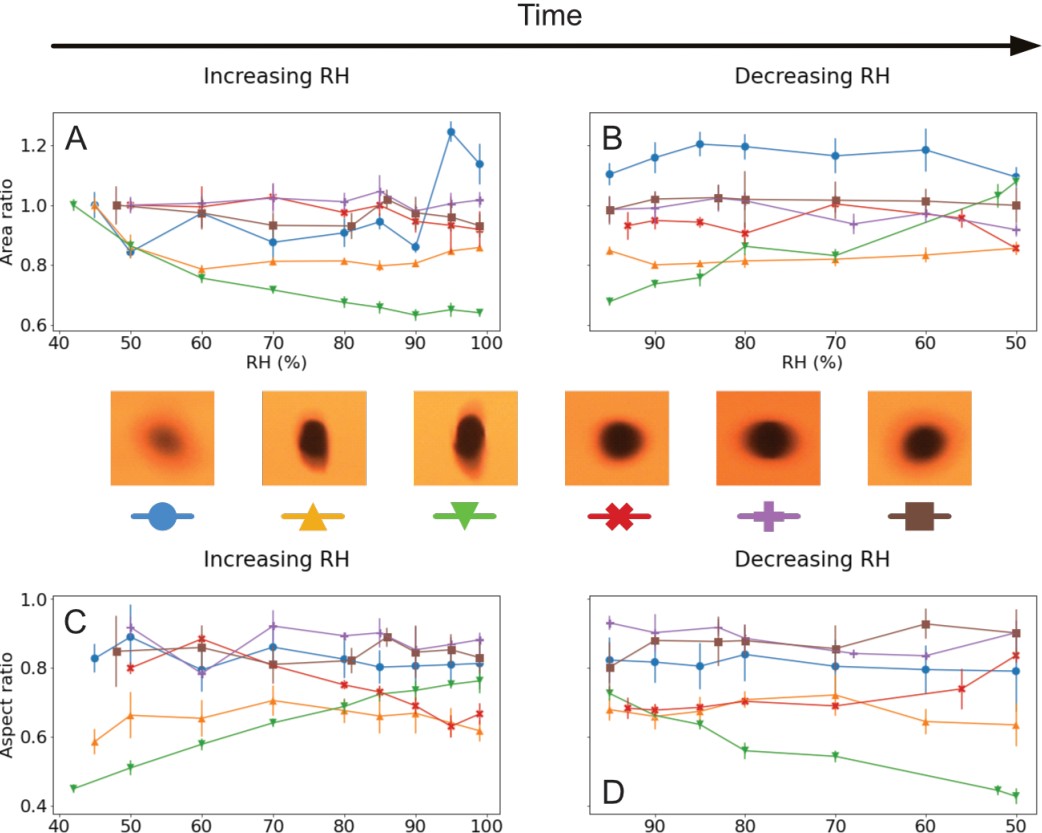

The results for visual size changes of the *Lilium orientalis* pollen while levitated, shown in Fig. 5, are somewhat inconclusive. Just one out of six experiments showed evidence of area increase, where the area increased beyond 10% with an RH increase from 45 to 99% RH. The other experiments show either no visible trend or even a significant decrease in measured area. The aspect ratio observations generally also did not show a conclusive trend. Initial ratios ranged between 0.45 and 0.92 demonstrating a large variance in observable shape among the grains. Once again, in cases where the aspect ratio was initially close to 1, generally less significant change was seen (including the grain that experienced significant area increase). The instance demonstrating the lowest aspect ratio was that which showed a particularly significant decrease in area with increased humidity. This grain also exhibited the most significant increase in aspect ratio with increasing humidity from 0.45 (at 42% RH) to 0.76 (99% RH) and back down to 0.43 (50% RH).

These results suggest that the measurement accuracy is hindered by the fact that the grains are being levitated freely. The significant but unexpected changes in observed area may be largely affected by their orientation while suspended, as there is no way to guarantee we are always looking at the same angle of the pollen grain. Once again, however, grains which are initially less circular show greater change in aspect ratio with humidity changes.

*Populus deltoides*

Since the *Populus deltoides* pollen was much more difficult to levitate stably than the *Lilium orientalis* pollen and often fell out of the trap, it was very difficult to collect a complete set of image data across all previous humidity increments. Therefore snapshots were only taken at room RH at 48% and maximum RH at 95% for increasing RH and then 90% and 50% RH in the decreasing direction. The pollen grain decreased by 22% of its original size when RH was increased to 95%. However, at the start of the decreasing RH direction, at 90% RH, it increased to 7% above its original size then to 9% when returned to 50% RH. Meanwhile the aspect ratio increased from 0.33 to 0.49 for increasing RH then decreased back to 0.32 when RH was decreased.

It should be noted that the images for this pollen grain even among the same RH increment displayed considerable variability (demonstrated in Fig. S5 in the SI) implying that the observed orientation was not fixed. Due to the instability of the levitated pollen grain, it was difficult to capture consistent snapshots of the grain even at constant RH. This can also be observed as the data points themselves for area and aspect ratio display considerable variance indicating a large error margin.

## 3.4 Summarised results

**Figure 6: Box-and-whisker summarising the change in area (left) and diameter (right) ratios for the RH interval 60-90% for each experiment group with overlaying swarmplot showing individual data points in red. (Lil. =** *Lilium orientalis*; **Pop. =** *Populus deltoides*; **Stat. = pollen grain was surface-fixed (static); Lev. = pollen grain was acoustically levitated.) The table below this reports the mean change in area and aspect ratios within in experiment group and the results of paired t-tests performed between area ratios at 60 and 90% RH, respectively.**

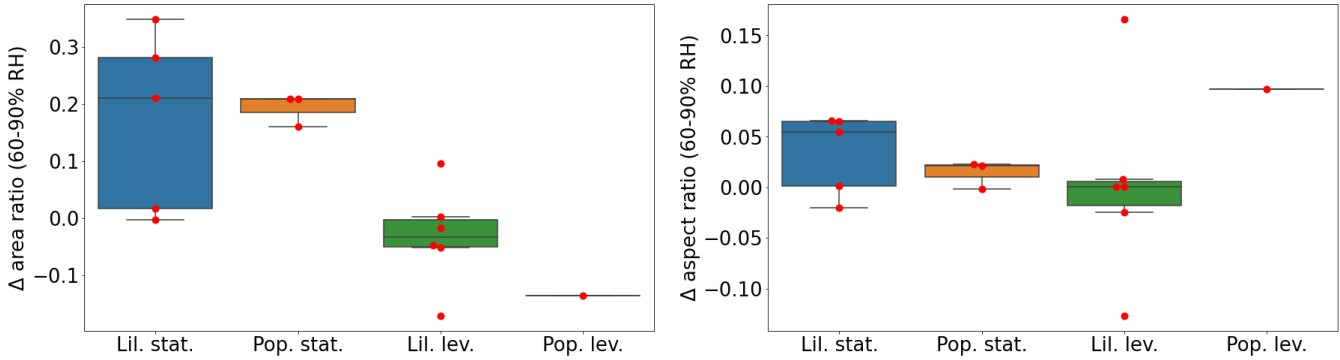

| | Mean change in area ratio between 60-90% RH | T statistic of paired t-test | p-value of paired t-test | Mean change in aspect ratio between 60-90% RH |
|---|---|---|---|---|
| *Lil.* **Stat.** | 0.17 | 2.4 | 0.07 | 0.033 |
| *Pop.* **Stat.** | 0.19 | 11.8 | 0.007 | 0.014 |
| *Lil.* **Lev.** | -0.03 | -0.9 | 0.4 | 0.004 |
| *Pop.* **Lev.** | -0.14 | - | - | 0.099 |

Fig. 6 summarises the change in area and aspect ratios between 60-95% RH for each experiment type, as estimated by the fitted linear regression models. The overlayed 'swarmplot' shows where the individual data points lie (in red), so as not to overlook that the number of samples composing the plots was varied and relatively sparse. Included is also a table summarising the mean values for area and aspect ratio change for each experiment type, and the results of a paired t-test compared between the estimated data points at 60 and 90% RH.

When calculated this way, the surface-fixed experiments showed an area increase of 17-19%. The results of the t-test demonstrated a significance level well within 0.05 for *Populus deltoides* while the *Lilium orientalis* only demonstrated significance at the 0.1 level. The proportional size change of the *Lilium orientalis* pollen may have been less evident when measured due to the larger size and different shape. Note that these two pollen species are each evolved for different pollination transport methods, which may affect their hygrscopic behaviour. The levitated pollen experiments could not demonstrate significant area change even at the 0.1 level. The aspect ratios also did not demonstrate signifiant change within the 0.1 level for any experiment type, however, as discussed previously, this seemed to be influenced by initial aspect ratio which is not accounted for here.

### 3.5 Comparison with previous methods

To put our results into context, we can compare them with findings from previous studies, presented in Table 1 in the Introduction section. Pope (2010) and Griffiths et al. (2012) measured a mass increase of 16-18% for various pollen species when RH was increased from 0 to 75%. When increased from 0 to 90% RH, Tang et al. (2019) and Chen et al. (2019) reported mass increases between 29-48%, an even further up to ~72% when increased to 95% RH.

However, the experiments in this study did not start at conditions of 0% RH. From observations for *Carya illinoinensis* (pecan) pollen by Tang et al. (2019), mass increases of 6.4%, 30.3% and ~72% were calculated when RH was increased from 0 to 60%, 90% and 95% RH, respectively. This means that there was a 22% mass increase between 60-90% RH and a 62% mass increase between 60-95%. These RH ranges are comparable to those tested in this study. Though, the reported literature values for mass increases above 90% RH show considerably large variation. This is predicted by Köhler theory which suggests that the hygroscopic increase increases rapidly at high RH (asymptotically towards 100%), and potentially the variation here is also greater here across different species. It may also suggest reduced accuracy for these methods under such high RH conditions. The RH range 60-90% was chosen for comparison in this study because the literature values appear more consistent.

Assuming the density of the pollen grain is constant and that the grains are nearly spherical, the relationship between area and mass increase ratios can be described by Eq. (2), where $m/m_0$ is the ratio of final mass over initial mass and $A/A_0$ is the ratio of final area over initial area:

$$\frac{m}{m_0} = \left(\frac{A}{A_0}\right)^{\frac{3}{2}} \tag{2}$$

The average area increases observed between 60-90% RH for the surface-fixed pollen experiments in this study were 17 and 19% for *Lilium orientalis* and *Populus deltoides*, respectively, which correspond to mass increases of 27 and 30%. This is a little higher than the 22% mass increase found between 60-90% RH for *Carya illinoinensis* pollen by Tang et al. (2019). Considering that the density of the pollen grain would also change while it is swelling with water, the relationship would more accurately be described by Eq. (3), where $\rho/\rho_0$ is the ratio of the final over the initial density of the pollen:

$$\frac{m}{m_0} = \left(\frac{A}{A_0}\right)^{\frac{3}{2}} \times \frac{\rho}{\rho_0} \tag{3}$$

It is difficult to determine exact values for pollen density due to their complex composition as a gamete from a biological organism, and it has not been shown how comparable estimates are between species. Taking an example from the literature, Sosnoskie et al. (2009) estimate the density of dry and fully hydrated *Amaranthus palmeri* pollen to be 1,435 and 1,218 kg m$^{-3}$, respectively. This corresponds to a 15% decrease in density when hydrated, or a $\rho/\rho_0$ ratio of 0.85. Applying this to equation 2, our area increases of 17 and 19% correspond to mass increases of 8 and 10%. Hence, corresponding mass increase values are considerably smaller when density change is taken into account.

*Carya illinoinensis* pollen grains generally range between 43-53 µm (Stone, 1963) and so fall into a size range above that of *Populus deltoides* but below that of *Lilium orientalis*, and thus may not be appropriate for comparison. Tang et al. (2019) also investigated one of the species in this study, *Populus deltoides*, and reported an average mass increase of 48% between 0 and 90% RH. For *Carya illinoinensis* the ratio between the mass increases between 60-90% and 0-90% RH was 0.74. Assuming the same ratio of *Populus deltoides* this would mean a mass increase of ~36% between 60-90% RH. By this comparison, the results from our study would appear to underestimate hygroscopic growth. However, the mass increases calculated by Eq. (2), without considering density changes, produce results that seem plausible in the context of the literature examples discussed here. Table 3 summarises these comparisons between the present study and literature examples from Tang et al. (2019).

**Table 3: Comparison of the method in this study and vapour sorption analyser method reported in literature, including calculated mass increase values for increasing RH between 60-90% for surface-fixed pollen experiments in this study with reported literature values. The ranges quoted for the examples in this study take into account that the density of the pollen grain may change with hygroscopic swelling. Advantages and disadvantages for each method are also summarised.**

| Reference | Method | Pollen species | RH interval | % mass increase | Advantages | Disadvantages |
|---|---|---|---|---|---|---|
| This study | Macroscope (& acoustic levitator) | *Lilium orientalis* | 60-90% | 8-27% | Possible to investigate pollen shape dynamics as well as clusters of particles. Lower cost equipment and versatile. | Sensitive to experiment lighting, particle orientation and instability when levitated. Difficult to obtain good quality quantitative results. |
| | | *Populus deltoides* | 60-90% | 10-30% | | |
| Tang et al. (2019) | Vapour sorption analyser | *Carya illinoinensis* | 60-90% | 23% | Currently more robust with better measurement accuracy. | Expensive equipment. Not possible to investigate particle shape. |
| | | *Populus deltoides* | 60-90% | ~36% | | |


However, directly comparing area or volume increase with mass increase does present some potential issues that should be considered. Pollen grains are rarely perfectly spherical, and shape varies across and, to an extent, among species. The grains have features in the exine (outer surface) such as apertures (thinner sections of wall where the pollen tube is designed to break through) which make grains somewhat asymmetrical (Linder and Ferguson, 1985). Under increased humidity conditions it
may be over-simplistic to assume that the grains would expand as a uniform sphere in all directions. Volume increase to accommodate the absorbed water may instead occur as an unfolding or inflating mode at places where the surface is creased or deflated, as opposed to an overall uniform grain expansion. Visual examples of this can be seen in ESEM (environmental scanning electron microscope) images presented previously by Pope (2010) and Griffiths (2012) and the mechanics of hygroscopic pollen inflation are studied by Božič and Šiber (2022). Therefore, comparing volume and mass increase ratios
may require appropriate consideration of such complexities. It may even be useful to consider alternative metrics to relate pollen state under given RH conditions to CCN ability, such as expansion of wetted surface area, if such attributes can be reliably measured.

The advantages that this method offers, in contrast to previous mass-measuring methods, include the ability to visualise directly particle behaviour and measure aspect ratios, providing quantitative information regarding the dynamic shape of the particle,
throughout changing experiment conditions. We have reported attempts at such measurements here for the first time, showing evidence of increased circularity - and by assumption sphericity - with increasing RH.

**4 Conclusions**

This work has, for the first time, explored the hygroscopic size change of pollen grains using acoustic levitation and visual imaging. Results have been compared with previous methods which measured mass change. Pollen was suspended by acoustic
levitation, allowing experiments investigating hygroscopicity in the true aerosol phase. The varying stability of the levitation in some cases, particularly for single smaller ($< 40$ μm) pollen grains, presented some difficulty. The real-time visual imaging of the levitated grains throughout experiments provides a valuable opportunity to understand the response of pollen grains under different conditions such as, but not limited to, varying humidity.

This method of observing pollen grains does have limitations, particularly when it comes to obtaining accurate quantitative
measurements. Though an unbiased computer program-based method was employed to extract measurements for particle area and diameters, it is evident that factors such as particle orientation and lighting conditions are important to consider. While lighting conditions can be kept constant for the course of an experiment, particle orientation is something we have little control over and affects the resulting image captured at any point in time. Fluctuations in orientation can give rise to considerable variation among snapshots of the same grain taken under constant conditions and presents an unknown that is difficult to
quantify. However, this technique does reveal details that are not possible to observe by previous techniques, such as the non-uniform mode of morphology change of pollen grains under changing humidity conditions. Future studies may benefit from higher resolution image capabilities or even multi-angle imaging to better elucidate particle orientation and consistent 3D changes in size and shape. If possible, it would be beneficial to achieve greater stability when trapping particles and record the power applied to quantify the acoustic pressure exerted on the particle accurately enough to measure mass changes as well.

In conclusion, this new method may not yet surpass previous mass change-quantifying methods in accuracy and leaves much to be improved upon. Nevertheless, there are novel measurements that can be made from the different perspective it provides, as well as various practical advantages such as lower cost and relative simplicity. We also suggest the importance of shape and surface area beyond simply the mass when considering potential CNN activity. We hope the methodology used in this work introduces a new possibility for further exploration involving the study of freely levitated aerosol or bio-aerosol particles
subjected to various environmental conditions.

**Code availability**

Code supporting this publication will be made openly available from the UBIRA eData repository https://doi.org/10.25500/edata.bham.00000923.

**Data availability**

Data supporting this publication are openly available from the UBIRA eData repository https://doi.org/10.25500/edata.bham.00000923.

**Author contribution**

The idea for this study was conceived and planned by FDP and CP. FDP, ARMK and CP provided general supervision and guidance. The practical experiments were carried out by SAM and AM. Analysis was completed and the first draft prepared by SAM. All authors contributed to the writing and editing of the final manuscript.

**Competing interests**

At least one of the authors is a member of the editorial board of Atmospheric Measurement Techniques.

**Acknowledgements**

SAM acknowledges the funding support of the NERC CENTA2 grant NE/S007350/1. AM acknowledges the support of the NERC SCENARIO DTP (NE/L002566/1). This research has also been supported by the Natural Environment Research Council (grant nos. NE/T00732X/1, NE/G000883/1 and NE/G019231/1) and the Royal Society (grant no. 2007/R2).

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
