# Peer review of "Acoustic levitation of pollen and visualisation of hygroscopic behaviour"

_EGUsphere, 2023_

## Referee Comment (RC4)

Review of: Acoustic levitation of pollen and visualisation of hygroscopic behaviour

by Sophie Mills, Adam Milsom, Christian Pfrang, Rob MacKenzie, and Francis Pope

Manuscript egusphere-2023-670 submitted to AMT

Pollen may act as atmospheric cloud condensation nuclei (CCN) as they are able to take up significant amounts of water vapor. The goal of the described experiments was to study hygroscopic swelling of pollen grains. The paper presents results from measurements with two types of pollen which were compared to results from other authors employing other techniques. Here, a common acoustic levitator was used where single pollen grains could be freely levitated and observed as well as recorded by a macroscope. It was combined with a humidity-controlled airflow. In contrast to previous measurements where the mass changes of the pollen were studied, the authors determined changes of area and diameter. This technique is new fur such types of measurements and the application for large primary biological particles such as pollen. The paper is very well written and easy to follow.

The advantage of this method is the possibility to directly observe the development of pollen sizes and shapes under varying conditions. However, the authors themselves claimed that their method is less accuarate than methods involving mass changes and requires modifications for further use.

Nevertheless, I think that the paper is worth publication after revision.

Major issues:

Figure 2: Better show only cutouts of these pictures with the pollen grains.

Table 2: The quality of the front-lit images seems much better. What is the reason and why did you use mainly back-lit?

The quality of the photos is rather low (Figure 2 and Table 2). I am wondering whether you do not have any images with better resolution.

Section 2.4 and Figures 3 to 6: Could you explain what exacty is meant by area ratio, as it was done for the diameter ratio?

To demonstrate the changes of pollen shapes during growth, I suggest to show images at different relative humidities. The introduction of a parameter describing the shape variation would be helpful.

The authors stated that the pollen grains become more spherical when they take up water vapor, the aspect ratio would be a good parameter to describe this.

Could you estimate the error (standard deviation) of their results so that the reader could better value them?

In line 293, the authors state that "These results suggest that the measurement accuracy is hindered by the fact that the grains are being levitated freely." What are the arguments that the advantages of freely leviated pollen prevail the low accuracy of the measurements?

Minor issues:

Line 223: This must be Figure 2 instead of Figure 3.

Line 144: remove "below"

Line 159: … accordingly, therefore, it is consistent only …

Line 195: remove the comma

Lines 223 – 225: the mentioned blue and orange lines are hardly visible

Lines 108,176,233,240,241,279,314, 338, 349, 357: comma before respectively

Line 344: nearly spherical

Lines 241/242: Is this sentence correct with the use of "yet"?

Line 245: Put "too" at the end of the sentence with a comma.

---

## Author Comment (AC3)

**Response to the reviewers for the article:**

**Acoustic levitation of pollen and visualisation of hygroscopic behaviour**

We thank the reviewers for their valuable time taken to carefully review this manuscript, their inputs have much improved the paper. We address all the points raised and justify our view that this work should be published.

**Reviewer 1 comments**

The authors presented results of pollen hygroscopic behaviour obtained from an acoustic levitation technique. From the two pollen grains study, using either deposited method or levitation method, area and diameter changes were presented, and compared with results from previous studies that were mainly done with mass measurements. The authors concluded that although the direct imaging methods might suffer from stability of the levitated pollen grains, they offer direct observation on the shape dynamics of the pollen grains during their hygroscopic growth. The technique itself is quite interesting in measuring hygroscopic behaviours of large particles such as pollen grains. The manuscript is also well written. I do, however, have a major concern on how to present the observed results, and would recommend Major Revision before publication

We thank the referee for their thorough reading of the paper. We agree with the referee that the technique at present does not yet surpass existing methods (e.g. EDB) for measuring hygroscopic behaviour of pollen. However, we believe the study demonstrates significant potential to provide useful new information on pollen, and potential also other large biochemical structures important in the atmosphere e.g. fungal spores etc. In common with other new techniques, the method still has aspects that can be improved upon with further technical iterations. We believe the paper will be of interest to the readers of AMT, and that the publication will encourage the field to iterate upon the initial 'version 1.0' to advance it to 'version 2.0' that can potentially rival the EDB, and other techniques.

Main:

The authors claimed that the main advantage of the acoustic levitation technique is that it can provide direct imaging on the shape change of the pollen grain during hygroscopic growth or shrinkage. Yet, there is little information on such an advantage, i.e., either images at different RH for the same pollen grain, or a parameter to show the shape "factor".

Please see the responses below on the aspect ratio measurement which caused some confusion and we have now clarified. This is the best measurement we could provide for the particle shape in this situation.

For the reader's interest and better understanding, we have added two Figures S4 and S5 to the Supporting Information which show images for surface-fixed *Lilium orientalis* and levitated *Populus*

*deltoides* across the humidity ranges. All images used for this study, as well as relevant code and datasets are available in the repository stated at the end of the article: https://doi.org/10.25500/edata.bham.00000923.

Please see additions to the manuscript:

*Ln 217-220: "Table 2 shows representative snapshots taken for each pollen type and experiment setup to show general image quality and the variation observed among them. Further images showing examples of the pollen grains across the relative humidity range, as well as the program-fitted contour and ellipse used for measurements, can be found in Figures S4 and S5 in the SI. All experiment images used in this study, as well as relevant code and datasets, can be found in the data repository in the Data Availability section."*

*Ln 236-237: "Table 2. Representative snapshots taken for each pollen type and experiment setup showing the image quality achieved as well as the variation across different instances. Further snapshots showing example pollen grains across the relative humidity range can be found in Figures S4 and S5 in the SI."*

*Ln 251: "The results for the Lilium orientalis pollen fixed to the levitation chamber window and under back-lit conditions are shown in Fig. 3. One pollen grain, the blue line with circle markers in Fig. 3 (further images presented in Fig. S4 in the SI), increased in silhouette area to just over 20% between 70-96% RH."*

*Ln 326: "It should be noted that the images for this pollen grain even among the same RH increment displayed considerable variability (demonstrated in Fig. S5 in the SI) implying that the observed orientation was not fixed."*

Related to the point above, in addition to the "area ratio" and "diameter ratio", would the aspect ratio be a good parameter to show that the pollen grains are becoming more and more spherical after taking up more water, as stated in the text?

We apologise for the confusion, what we meant by "diameter ratio" is in fact "aspect ratio" and serves as a measure of circularity/sphericity. We have changed all instances of 'diameter ratio' in the manuscript to 'aspect ratio' to make this clear to readers as well as changing the appropriate paragraph in the methods section that now reads as follows:

*Ln 162-171: "Pixel area measurements have been converted into average area increase ratios (averaged over 5 repeat snapshot images) relative to the initial size measured at ambient RH. The ellipse-fitted diameter measurements do not necessarily correspond with 'polar' and 'equatorial' axes defined by general pollen terminology, as this cannot be determined with certainty from the images. However, we used the ellipse diameter measurements to calculate an average aspect ratio to evaluate change in shape. The aspect ratio was defined by the following Eq. (1):*

$$Aspect\ ratio\ =\ \frac{smaller\ ("equatorial")\ diameter}{larger\ ("polar")\ diameter} \qquad (1)$$

*This resulted in values between 0 and 1, with higher values implying more circularity (and thus likely higher sphericity of the whole pollen grain as well)."*

It seems from the surface-fixed results that front lit results were clearer (Table 2 and Figure 3). How come it was not used in the acoustic levitation method later on?

We considered the back-lit silhouette generally produced a more desirable contrast between grain and background. Markings on the chamber window were visible during the front-lit levitation experiments, causing difficulties for the program when finding the correct object to contour. When first experimenting with different lighting angles, there did not seem to be significant difference in quality, i.e. a more crisp outline for front-lighting, in general. Please note that the front-lit images presented here represent just one instance, also when the grains were stationary, so it is not appropriate to make the comparison with all the other back-lit images presented in terms of general quality. The decision was made to continue with the back-lit setup mainly for the reason that the

whole picture generally seemed much clearer with only the pollen grain(s) visible and no obscuring background objects. We have just decided to include an example of an initial experiment performed with the front-lit setup for completeness, interest to the reader, and so that it may not be discarded as an option if others choose to continue this work in the future.

The following has been changed/added in the manuscript:

*Ln 222-232: "From initial experiments, back-lighting was chosen as a default to proceed with measurements as it resulted in a generally crisper and more easily definable silhouette. Importantly, it also produced a cleaner background devoid of other features that would interfere with the Canny edge detection and contour identification. (Front-lighting resulted in a less-pristine background due to other features picked up from the glass chamber window.)*

*In the following results, we include for comparison an example where two pollen grains were observed under front-lit conditions. After post-experiment analysis, it appeared that back-lit conditions may not necessarily achieve more consistent results than the equivalent front-lit experiment. The decision to proceed with back-lighting was made based on observations at the time for the reasons above, in particular with the automated method to distinguish pollen grains from the image background in mind. However, the method could be refined to overcome interference of background features and it may be that front-lit experiments are worth testing again in the future, as this may produce more consistent images that vary less with light brightness and contrast settings."*

Minor:

L220-230: Fig. 2 should be Fig. 3?

We thank the reviewer for catching this, the relevant instances have now been updated in the manuscript.

Could the authors comment on how potential lateral motion during acoustic levitation would affect the determination of area and diameter ratios?

We have commented on general issues of motion and instability but cannot isolate specifically lateral motion. In our method we take 5 repeat snapshots for each and take average measurements to minimise such effects. This was not clearly explained in the methods section, so we have made appropriate changes to lines 162-166 to address this.

Additionally, we have added error bars to the plots based on the standard deviation between the 5 snapshots in each case. These error bars demonstrate that the discrepancy between snapshots taken in quick succession is small. This suggests that the lateral motion within the short timescale of a few seconds is not significant, or at least does not greatly affect the measurements. Rather, it is generally the changes that occur over a greater timescale and as the pollen grains are undergoing changes in mass, position within the acoustic field and orientation due to the environment.

*Ln 162-166: "Pixel area measurements have been converted into average area increase ratios (averaged over 5 repeat snapshot images) relative to the initial size measured at ambient RH. The ellipse-fitted diameter measurements do not necessarily correspond with 'polar' and 'equatorial' axes defined by general pollen terminology, as this cannot be determined with certainty from the images. However, we used the ellipse diameter measurements to calculate an average aspect ratio to evaluate change in shape.*

Figures 3-6 need further modification to font size bigger.

Figures 3-6 font sizes have been adjusted.

Is there a way to obtain some estimates of the uncertainties for the area ratios and diameter ratios, such that readers can appreciate what changes can be understood as significant?

We have now added error bars to Figs 3-5 which are from the standard deviation of the 5 repeats used for averaged values. The following sentence has also been added to the methods section:

*Ln 171-172: "The standard deviation was calculated across the 5 repeats for each averaged value of area and aspect ratios and is presented as error bars in the figures in Results and Discussion sections 3.2 and 3.3."*

How was the diameter ratio defined (i.e., normalized to what)? Why is it always less than unity?

Please see point above on aspect ratio and additions to the manuscript methods section.

**Review 2 comments**

Mills et al. present an acoustic levitation technique and the results of hygroscopic water uptake experiments of two types of pollen grains, from Oriental Lily and from Cottonwood (roughly 30 to 100 microns in diameter). The commercial acoustic levitator is coupled to a humidity-controlled air flow, and the RH used ranged from dry to 95%. While the technique and its application to primary biological particles are interesting to the atmospheric science community.

However, there are some issues with the framing of the study. More precision and substantiation are needed in the introductory sketch of the state of the science. Further, the technique is not able to measure water uptake by pollen grains with enough precision and enough repeatability for the results to be conclusive. The technique does not represent an improvement to existing methods, and therefore the publication of this technique is in doubt.

Thank you for your careful and thorough comments on this manuscript. We attempt to address your specific points below. In general, we fully acknowledge that our results here do not surpass those measured by conventional methods. However, we present here an alternative method that does present attractive advantages in terms of cost and availability and has the potential to provide novel visual data that previous methods cannot offer. We do not present a polished alternative method that is ready to use as a benchmark, however we feel it is important to report what we have found in this pilot study so that others in the community to facilitate future progress and, in particular, where experiment engineering advancements may have to be considered.

Comments on the introductory text

The claim that discussion of pollen as CCN in the literature is sustained or increasing should be more carefully supported. The importance of pollen in the atmosphere is not limited to impacts on CCN, perhaps other impacts should be emphasized. (Line 8-9, Abstract ("Pollen are hygroscopic and so have the potential to act as cloud condensation nuclei (CCN) in the atmosphere. This could have yet uncertain implications for cloud processes and climate."), line 61-62, Introduction ("While there have been increasing discussions in the literature postulating the significance pollen may have for atmospheric cloud processes and climate, …")).

The wording that suggests there have been 'increasing' discussions of pollen as CNN has been adjusted, since this is perhaps misleading. There have been discussions, in the references quoted, but not necessarily increasingly so in recent years, as you have pointed out.

Indeed, we agree that there are other important implications of pollen in the atmosphere on aspects such as health and biodiversity. Hygroscopicity may also have implications for successful gene transmission and biodiversity, and perhaps even public health. The update of water by pollen alters the size, shape and mass of pollen grains and will affect the aerodynamics of pollen within the atmosphere. With this in mind, we have altered the following lines of the manuscript and additional references added (Matthews et al., 2023; Hughes et al., 2020; Cecchi et al., 2021).

*Ln 62-68: "While there have been discussions in the literature postulating the significance pollen may have for atmospheric cloud processes and climate, there is need for further experiments to measure and better understand the hygroscopic behaviour of pollen and other biological particles (Möhler et al., 2007; Després et al., 2012; Sun & Ariya, 2006). Pollen hygroscopicity may also have important implications for airborne transmission to new areas, and thus the colonisation and biodiversity of various plant species. Additionally, it is important to consider for human health, particularly since the release of respirable-sized subpollen particles from pollen grains can be modulated by humidity conditions (Matthews et al., 2023) and stimulated by events such as thunderstorms (Hughes et al., 2020; Cecchi et al., 2021)."*

*Ln 8-9: Abstract "Pollen are hygroscopic and so have the potential to act as cloud condensation nuclei (CCN) in the atmosphere. This could have yet uncertain implications for cloud processes and climate, as well as plant biodiversity and human health."*

The impact of giant pollen CCN on cloud droplet number or supersaturation should be considered in more detail by the authors before being suggested. (Lines 55-59, Introduction)

The following paragraph has been updated in the manuscript introduction to address this:

*"Though relatively small in number concentration, the large size of pollen grains mean that they can act as coalescence embryos, also known as giant cloud condensation nuclei (GCCN). These GCCNs have been shown within atmospheric models to have a disproportionate contribution to the development of precipitation within clouds (Johnson, 1982; Cotton and Yueter, 2009). They can nucleate cloud droplets at lower supersaturations than other smaller aerosols and facilitate precipitation more rapidly (Després et al., 2012; Pope, 2010; Posselt & Lohmann, 2008), and particularly affect precipitation in clouds of high droplet concentration, i.e. polluted as opposed to pristine clouds, even at concentrations as low as $10^{-3}$ cm$^{-3}$ (Möhler et al., 2007)."*

The claim that acoustic levitation has progressed significantly in recent years, and that this is a good way to study aerosol, should be substantiated by citation to recent papers (Lines 114-116).

We thank the reviewer for the need for substantiation of this statement. It is the application of acoustic levitation that has progressed in recent years. We have edited this statement in the text for clarity and with examples as follows:

*"The application of acoustic levitation has progressed significantly in recent years. For example, acoustic levitation has been coupled with techniques such as Raman spectroscopy (Milsom et al, 2021; Pfrang et al., 2017) and X-ray scattering experiments (Milsom et al., 2021; Milsom et al., 2022; Pfrang et al., 2017; Seddon et al.; 2016). Acoustic levitation presents opportunities to study substances in contactless and container-less conditions, i.e. in the true aerosol phase."*

The assumption that pollen grains can restructure when humidified should be substantiated and discussed here in the context of the findings presented, as the implications of pollen restructuring are broad (Line 250)

While visual evidence can be found in Pope (2010) and Griffiths et al. (2012) and a modelling study of the mechanics can be found in Božič and Šiber (2022), the mechanics of pollen grain unfolding and restructuring when humidified are still poorly understood. Hence we think it important to develop methods that can empirically study these visual aspects. We have altered the following sections of the manuscript:

*Ln 77-84: "Pollen grains are complex biological structures with varying sizes, shapes and surface features across species. Previous methods have focused on the measurement of mass increase with hydration and have not considered the simultaneous changes that may occur in volume and density. For example, Božič and Šiber (2022) explore the mechanics of pollen grain unfolding, swelling and bursting under humidity changes, which is largely dictated by the soft apertures, or pores, found on the exine (outer surface). The number and size of pores can vary across different pollen taxa but the mechanics involving them are not well understood. Alongside modelling studies such as Božič and Šiber (2022), empirical studies which shed light on the visual changes that occur on the pollen grain surface would bring value. With these considered, we may better understand the complexities of pollen hygroscopic behaviour and how it should be modelled in the atmosphere."*

*Ln 419-421: "Visual examples of this can be seen in ESEM (environmental scanning electron microscope) images presented previously by Pope (2010) and Griffiths (2012) and the mechanics of hygroscopic pollen inflation are studied by Božič and Šiber (2022)."*

Comments on the technique

The accuracy of the measurements is low, as noted by the authors and as evident in the spread in the area and diameter ratios displayed in Figure 6 (see, e.g., line 247-249 ("While these results show some evidence of the hygroscopic size increase we had expected with increasing RH, they also suggest apparent inaccuracy and lack of consistency which must be considered. This variability may be due to limitations of the method in terms of imaging capabilities."), Line 271 ("yet the change may not be considered conclusive."); line 275 ("it would be necessary to conduct more experiment instances before making conclusive assertions from these results"); line 284 ("The results for visual size changes of the *Lilium orientalis* pollen while levitated are somewhat inconclusive."); line 287 ("generally also did not show a conclusive trend."); line 293 ("These results suggest that the measurement accuracy is hindered by the fact that the grains are being levitated freely."); Line 305 ("It should be noted that the images for this pollen grain even among the same RH increment displayed considerable variability, implying that the observed orientation was not fixed."); lines 306-308 ("Due to the instability of the levitated pollen grain, it was difficult to capture consistent snapshots of the grain even at constant RH. This can also be observed as the data points themselves for area and diameter ratio display considerable variance indicating a large error margin.")).

We are pleased that the reviewer has noticed the care we have taken to caveat our results and present them honestly. We fully recognise the results do not surpass existing methods in terms of robustness or accuracy. However, we argue that there is potential to provide useful new visual information and that the method has many aspects that can be improved upon if taken further. We argue that certain engineering advancements may be more crucial than increased experimental and statistical rigour when it comes to improving results and so this study is important to highlight this.

This is, to our knowledge, the first study introducing the acoustic levitation of pollen grains and we believe it is informative to the wider scientific community. This study documents the current capabilities, limitations, and potential of an initial 'version 1.0'. We think it important to report these findings so that others can be informed and take these considerations into account, and this will facilitate productive advancements to 'version 2.0' and so on in the future.

It is assumed in the calculations that the grains are always the same distance from the camera, even though migration toward and away from the camera has been noted. The uncertainty in size due to positioning should be quantified. (Line 293-295 ("The significant but unexpected changes in observed area may be largely affected by their orientation while suspended, as there is no way to guarantee we are always looking at the same angle of the pollen grain.")).

We acknowledge this is a concern. We have commented on general issues of motion and instability but cannot isolate specifically lateral motion or quantify distance from the camera. In our method we take 5 repeat snapshots for each increment and take average measurements to minimise such effects. This was not clearly explained in the methods section, so we made appropriate changes to lines 162-166 to address this.

Additionally, we have added error bars to the plots based on the standard deviation between the 5 snapshots in each case. These error bars demonstrate that the discrepancy between snapshots taken in quick succession is small. This suggests that the lateral motion within the short timescale of a few seconds is not significant, or at least does not greatly affect the measurements. Rather, it is generally the changes that occur over a greater timescale and as the pollen grains are undergoing changes in mass, position within the acoustic field and orientation due to the environment.

*Ln 162-166: "Pixel area measurements have been converted into average area increase ratios (averaged over 5 repeat snapshot images) relative to the initial size measured at ambient RH. The ellipse-fitted diameter measurements do not necessarily correspond with 'polar' and 'equatorial' axes defined by general pollen terminology, as this cannot be determined with certainty from the images. However, we used the ellipse diameter measurements to calculate an average aspect ratio to evaluate change in shape.*

Comments on results

Perhaps more data should be collected. (Line 299 ("it was very difficult to collect a complete set of image data across all previous humidity increments")).

We agree this would have been desirable, given unlimited time constraints and funding, but argue that even if more data had been collected, it would not have overcome the current engineering limitations. We think this is more of an issue to be solved by engineering advancements than simply more experiments and brute statistical force. Therefore, we believe it is important to document our current experiences with this new approach to pollen measurements to inform future progress, by highlighting the advantages, as well as the difficulties and limitations encountered.

The hygroscopicity derived from the reported results falls within the wide range for pollen reported by the literature. However, hygroscopicity is an intensive property of fully dissolved molecules. The correct term to apply in pollen studies, assuming some insoluble fraction, is the "apparent hygroscopicity." The wet pollen grain is a mixture of soluble molecules and an insoluble part.

Yes, indeed the ranges reported across the literature sources are considerably wide, suggesting that the conventional techniques also had their own uncertainties. We acknowledge that the values derived from the calculations in the later part of the discussion for comparison with literature values will have considerable associated uncertainties. We by no means suggest using this method as a benchmark to confirm such values, but the calculations are simply a demonstration of how visual data from the method we present can be compared with the mass information from other methods.

As for the point on apparent hygroscopicity, thank you for picking up on this, we have added the following sentence to the introduction in the manuscript to clarify.

*Ln 54-55: "Note that when we refer to hygroscopicity of pollen it should more correctly be 'apparent hygroscopicity', since pollen grains are a mixture of soluble and insoluble components. For brevity we use hygroscopicity henceforth in the paper."*

The figure text should be roughly the same size as the caption text after the figures are resized to their final publication-ready dimensions. As submitted, the figure fonts are about 50% as large as the figure caption font, meaning that the font size should be doubled and the plot area reduced accordingly. Some guidelines also suggest that data symbols appear similar in size to the fonts.

Thank you for pointing this out to us. We have taken note of this and made sure to enlarge the text where it was too small and hope it is now sufficiently legible.

**Reviewer 3 comments**

Mills et al. reported an interesting study that echos Robert Brown's experiment 200 years ago. Yet, in the present case, the instability of small pollen particles is a technical issue to be addressed. The scope of the present study fits with that of Atmos. Meas. Tech. The manuscript is well written and easy to follow. I recommend the publication of this article after the following technical concerns are satisfactorily addressed.

Technical:

To what extent does the acoustic field affect the thermodynamic properties of the air surrounding the levitated particles? When acoustic standing wave is formed inside the levitation chamber, I would assume that the pressure and density of the air near the levitated particles differ from that of ambient air. Yet the relative humidity (RH) was measured outside the acoustic field. This gives rise to two technique questions: First, does the measured ambient RH accurately reflect the true RH near the surface of the levitated particles in the acoustic field? Second, more importantly, how does the acoustic field affect the mass transport of airborne water molecules, such as their flux from the ambience to the surface of levitated pollen particles.

Though we cannot quantify the extent that the acoustic field affects the thermodynamic properties of the air surrounding the droplet, it is known that acoustic streaming could affect the kinetics of evaporation processes in liquid droplets [Zang et al, 2017; Yarin et al, 2002]. Acoustic streaming refers to solvent-enriched vortices created near a liquid particle surface driven by acoustic waves [Zang et al, 2017]. To what extent this is happening around a solid pollen particle, we do not know. However, the fact that we are constantly flowing a gas past the levitated particle ensures that we minimise this [Yarin et al, 2002].

Zang et al, 2017: http://dx.doi.org/10.1016/j.cis.2017.03.003

Yarin et al, 2002: https://doi.org/10.1016/S0142-727X(02)00142-X

So, to address the two points: 1) the measured RH is likely an accurate representation of the RH near the pollen surface due to the constant replenishment of the near-surface gas phase and inhibition of

acoustic streaming; 2) the acoustic field is likely to affect the kinetics of evaporation – however, we are interested in hygroscopicity in this study, which is a thermodynamic property and therefore measured at equilibrium.

It is indeed difficult to measure the pollens' size accurately when they are vibrating. I encourage the authors to perform statistically analysis on the pollens' image in a more systematic manner. For example, one may calculate the pixel values (i.e., brightness or darkness of a pixel, hereafter, P) and then plot the pixel value P as a function of pixels' distance to the geometric center of the pollen (hereafter r). This P(r) distribution function may comprise the neccesary information to quantify (or, better, filter out) the blurriness owing to the vibration of pollen particles. Next, one may fit the P(r) function to invert a length scale parameter (hereafter L), a length which can be related to the known pollen size (hereafter, dp). For example, one may establish this L versus dp relationship at dry condition. This relationship can then serve as a calibration curve to invert the dp from L during the vary RH experiments. Try it out and see whether the uncertainty could be better constrained.

This is an interesting idea but, regretfully, we believe this is beyond the scope of this work.

Presentation:

The author may consider placing scale bars next to the pollens' image.

The method using the macroscope did not allow for us to produce absolute size measurements for the pollen grain images (this should not matter since all results are discussed in terms of relative size). However, we did make average size measurements for pollen grains from each of the samples. These are reported in section 2.1 (shown below) and alongside Fig. S1 in the SI.

*"Pollen grains from the samples used in this study (see Fig. S1 in the SI for microscope images) were measured using a Nikon SPZ1000 stereo microscope equipped with a Nikon DS-Fi1 camera and controlled by NIS-Elements acquisition software at the Birmingham Advanced Light Microscopy (BALM) facility at the University of Birmingham. The Lilium orientalis pollen from fresh flowers was measured to have polar and equatorial diameters ranging between 92-122 μm and 43-67 μm and means of 108.6 μm and 50.9 μm, respectively. The Populus deltoides pollen used for this study was measured to have diameters across multiple axes ranging between 18 and 32 μm, with a mean of 24.8 μm. There is however considerable variation in size and shape across even pollen grains of the same taxa since these are gametes produced from living organisms with complex biochemistry and physiology."*

Minor:

Line 29-39: "PBAPs have been found to constitute almost 25% of insoluble aerosol particles..." Is the 25% a mass fraction?

Apologies for the confusion, this is almost 25% by number concentration. The sentence has been corrected to the following to avoid confusion:

*"PBAPs have been found to constitute up to 25% of insoluble aerosol particle number concentrations with diameters greater than 0.2 μm in cloud water (Matthias-Maser et al., 2000) …"*

Line 38-47: This paragraph discusses the size range and the aerodynamic properties of pollens. I am curious about the size parameters mentioned here. Are they aerodynamic diameters or phyiscal size? Please specify.

*This was referring to physical size. Apologies, there may have been some confusion since the references we were referring to were ones to be found within that cited. We have updated this to refer to the relevant literature more specifically.*

*"Pollen particles are among the largest in physical size of PBAPs, generally measuring between 10 and 100 µm in diameter (see, e.g., Després et al., 2012), yet vary considerably in size and shape between and even within species."*

Line 48: "altitude of 3000m".  3km is better

*Thank you, we have adjusted this.*

Line 48-49: "considerable lengths of time with favorable meteorology" is vague. How long extactly do pollens reside in the troposphere? How is their lifetime compared with the characteristic time of cloud processes?

*We have updated this paragraph with more details and relevant references to be more specific.*

*"Airborne pollen concentrations are measured to be anywhere between 10-1000 grains $m^{-3}$ (Després et al., 2012), but can reach $10^4$-$10^6$ grains $m^{-3}$, particularly in the Northern hemisphere spring months of April and May when many deciduous trees peak in pollen emissions (Steiner et al., 2015; Gregory, 1978). Pollen grains have been shown to reach altitudes of 3 km and above in the atmosphere (Diehl et al., 2001) and travel horizontal distances of up to ~$10^3$ km (Sofiev et al., 2006; Hjelmroos, 1992; Sack, 1949). Individual small pollen grains can reside in the atmosphere for a few (generally 2-4) days (Sofiev et al., 2006), though pollen seasons can persist with high average concentrations (~$10^4$ grains $m^{-3}$) over many months, fluctuating with locality, season, time of day and meteorology (Gregory, 1928)."*

Line 55 "relatively small in number concentration" is again not concrete enough. What is the typical number concentration?

*The typical number concentrations with their typical fluctuations have been discussed in detail previously in the introduction (i.e. in amended paragraph for previous comment). While concentrations vary greatly, it established that number concentrations are generally lower than other PBAPs. Table 4 in Després et al. suggests a general number concentration difference of 2-3 orders of magnitude. The following has been added to the sentence to provide this specific detail for comparison:*

*"Though relatively small in number concentration (i.e. generally 10 $m^3$ compared to the $10^3$-$10^4$ $m^3$ number concentrations of fungal spores, bacteria and viral particles – see, e.g., Després et al.), ..."*

**Reviewer 4 comments**

Pollen may act as atmospheric cloud condensation nuclei (CCN) as they are able to take up significant amounts of water vapor. The goal of the described experiments was to study hygroscopic swelling of pollen grains. The paper presents results from measurements with two types of pollen which were compared to results from other authors employing other techniques. Here, a common acoustic levitator was used where single pollen grains could be freely levitated and observed as well as recorded by a macroscope. It was combined with a humidity-controlled airflow. In contrast to previous measurements where the mass changes of the pollen were studied, the authors determined changes of area and diameter. This technique is new for such types of measurements and the application for large primary biological particles such as pollen. The paper is very well written and easy to follow.

The advantage of this method is the possibility to directly observe the development of pollen sizes and shapes under varying conditions. However, the authors themselves claimed that their method is less accurate than methods involving mass changes and requires modifications for further use.

Nevertheless, I think that the paper is worth publication after revision.

Major issues:

Figure 2: Better show only cutouts of these pictures with the pollen grains.

This has been changed.

Table 2: The quality of the front-lit images seems much better. What is the reason and why did you use mainly back-lit?

We considered the back-lit silhouette generally produced a more desirable contrast between grain and background. Markings on the chamber window were visible during the front-lit levitation experiments, causing difficulties for the program when finding the correct object to contour. When first experimenting with different lighting angles, there did not seem to be significant difference in quality, i.e. a more crisp outline for front-lighting, in general. Please note that the front-lit images presented here represent just one instance, also when the grains were stationary, so it is not appropriate to make the comparison with all the other back-lit images presented in terms of general quality. The decision was made to continue with the back-lit setup mainly for the reason that the whole picture generally seemed much clearer with only the pollen grain(s) visible and no obscuring background objects. We have just decided to include an example of an initial experiment performed with the front-lit setup for completeness, interest to the reader, and so that it may not be discarded as an option if others choose to continue this work in the future.

The following has been changed/added in the manuscript:

*Ln 222-232: "From initial experiments, back-lighting was chosen as a default to proceed with measurements as it resulted in a generally crisper and more easily definable silhouette. Importantly, it also produced a cleaner background devoid of other features that would interfere with the Canny edge detection and contour identification. (Front-lighting resulted in a less-pristine background due to other features picked up from the glass chamber window.)*

*In the following results, we include for comparison an example where two pollen grains were observed under front-lit conditions. After post-experiment analysis, it appeared that back-lit conditions may not necessarily achieve more consistent results than the equivalent front-lit experiment. The decision to proceed with back-lighting was made based on observations at the time for the reasons above, in particular with the automated method to distinguish pollen grains from the image background in mind. However, the method could be refined to overcome interference of background features and it may be that front-lit experiments are worth testing again in the future, as this may produce more consistent images that vary less with light brightness and contrast settings."*

The quality of the photos is rather low (Figure 2 and Table 2). I am wondering whether you do not have any images with better resolution.

These are cutouts of the larger picture that the macroscope took. The pollen grains were relatively small within the whole picture captured. Unfortunately, due to the limitations of the macroscope, we do not have images of better resolution than this.

Section 2.4 and Figures 3 to 6: Could you explain what exactly is meant by area ratio, as it was done for the diameter ratio?

This has been done now:

*"Pixel area measurements have been converted into average area increase ratios (averaged over 5 repeat snapshot images) relative to the initial size measured at ambient RH (for each experiment), defined in Eq. (1):*

$$Area\ ratio\ = \frac{area\ within\ contour\ at\ given\ RH\ (pixels)}{area\ at\ initial\ RH\ (pixels)} \tag{1}$$

*"*

To demonstrate the changes of pollen shapes during growth, I suggest to show images at different relative humidities. The introduction of a parameter describing the shape variation would be helpful.

Please see response to Reviewer 1:

For the reader's interest and better understanding, we have added two Figures S4 and S5 to the Supporting Information which show images for surface-fixed *Lilium orientalis* and levitated *Populus deltoides* across the humidity ranges. All images used for this study, as well as relevant code and datasets are available in the repository stated at the end of the article: https://doi.org/10.25500/edata.bham.00000923.

Please see additions to the manuscript:

*Ln 217-220: "Table 2 shows representative snapshots taken for each pollen type and experiment setup to show general image quality and the variation observed among them. Further images showing examples of the pollen grains across the relative humidity range, as well as the program-fitted contour and ellipse used for measurements, can be found in Figures S4 and S5 in the SI. All experiment images used in this study, as well as relevant code and datasets, can be found in the data repository in the Data Availability section."*

The authors stated that the pollen grains become more spherical when they take up water vapor, the aspect ratio would be a good parameter to describe this.

Please see response to Reviewer 1.

We apologise for the confusion, what we meant by "diameter ratio" is in fact "aspect ratio" and serves as a measure of circularity/sphericity. We have changed all instances of 'diameter ratio' in the manuscript to 'aspect ratio' to make this clear to readers as well as changing the appropriate paragraph in the methods section that now reads as follows:

*Ln 162-171: "Pixel area measurements have been converted into average area increase ratios (averaged over 5 repeat snapshot images) relative to the initial size measured at ambient RH. The ellipse-fitted diameter measurements do not necessarily correspond with 'polar' and 'equatorial' axes defined by general pollen terminology, as this cannot be determined with certainty from the images. However, we used the ellipse diameter measurements to calculate an average aspect ratio to evaluate change in shape. The aspect ratio was defined by the following Eq. (1):*

$$Aspect\ ratio\ =\ \frac{smaller\ ("equatorial")\ diameter}{larger\ ("polar")\ diameter} \tag{1}$$

*This resulted in values between 0 and 1, with higher values implying more circularity (and thus likely higher sphericity of the whole pollen grain as well)."*

Could you estimate the error (standard deviation) of their results so that the reader could better value them?

This has now been done and error bars added to the plots as per other reviewers' comments.

In line 293, the authors state that "These results suggest that the measurement accuracy is hindered by the fact that the grains are being levitated freely." What are the arguments that the advantages of freely levitated pollen prevail the low accuracy of the measurements?

The main argument is that we can observe the behaviour of the pollen grains while in the free aerosol phase, i.e. without contact with a surface, so it is most realistic for aerosol particles suspended in the atmosphere. This is why we were interested in attempting this study. The results however show that this may not be as easy as initially thought with this method. The surface-fixed results demonstrate the potential that could be gained by visualising with the macroscope, if stability issues regarding the acoustic levitation of pollen can be improved upon in the future.

Minor issues:

Line 223: This must be Figure 2 instead of Figure 3.

We have changed the instance to what it should be "Fig. 3".

Line 144: remove "below"

This has been corrected.

Line 159: … accordingly, therefore, it is consistent only …

This has been corrected.

Line 195: remove the comma

This has been corrected.

Lines 223 – 225: the mentioned blue and orange lines are hardly visible

Confused Fig. 2 & 3?

Lines 108,176,233,240,241,279,314, 338, 349, 357: comma before respectively

These have been corrected.

Line 344: nearly spherical

This has been corrected.

Lines 241/242: Is this sentence correct with the use of "yet"?

Yes, I think so… because the reducing RH trend did not reverse that of increasing RH, yet, in contrast, stayed more or less consistent.

Line 245: Put "too" at the end of the sentence with a comma.

This has been corrected.

---

## Author Response (AR2)

Dear Editor,

Many thanks for accepting the paper. I've updated one missing reference in the bibliography, which was cited in the text but was missing in the bibliography.

Best Wishes,

Francis Pope